# Physicochemical Characterization of Natural Wollastonite and Calcite

**Murat Olgaç Kangal [1,\*], Gülay Bulut [1] and Onur Guven [2]**

[1]  Mineral Processing Engineering Department, Faculty of Mines, Istanbul Technical University, Maslak, 34469 Sarıyer/Istanbul, Turkey; gbulut@itu.edu.tr
[2]  Mining Engineering Department, Faculty of Engineering, Adana Alparslan Türkeş Science and Technology University, Balcalı, Çatalan Cd. 201/5, 01250 Sarıçam/Adana, Turkey; oguven@atu.edu.tr
[\*]  Correspondence: kangal@itu.edu.tr

**Abstract:** Wollastonite and calcite minerals are significant raw materials and are extensively used due to their unique properties. Wollastonite is used in plastics, paint, ceramics, paper, resins, and in construction as a substitution for asbestos due to its chemical stability, thermal resistivity, needle-like shape, and brightness. Calcite is one of the most used raw materials because of its low hardness, high alkalinity, sorptive properties, white and bright color. Wollastonite and calcite are two minerals found together in nature. The most common method used for separating these two minerals is flotation. In this study, the surface properties of pure mineral samples were investigated. The pH profiles of both minerals were obtained by measuring the surface charge of particles followed by the measurement of the zeta potential in different collector concentrations. The wettability of minerals was examined by measuring their contact angles.

**Keywords:** wollastonite; calcite; zeta potential; contact angle

---

## 1. Introduction

Wollastonite is an industrial mineral that can also be defined as calcium metasilicate ($CaSiO_3$) formed by the metamorphism of siliceous limestone at temperatures around 450 °C or over and occurs in regionally metamorphosed high-grade rocks and near igneous contact zones [1]. It is used for ceramics, paints, plastics, and as a substitute for asbestos. As extensively explained in a recent study, the aspect ratio is the most decisive point for its usage. In this manner, while it is suitable for reinforcing thermoplastics and thermoset polymer compounds and as a substitute for asbestos at high aspect ratios, such as in the range of 15:1 to 20:1, wollastonite is chosen for usages in ceramics, metallurgical fluxes, or simple filler and coating at low aspect ratios within the range of 1:3 to 1:5 [2].

Over the last decade, due to the depletion of coarse sized and high-grade ores, the enrichment of low-grade and fine-sized ores has become inevitable. Most of the studies have been devoted to finding out the underlying mechanisms to beneficiate these kinds of ores [1–6]. Although the possible enrichment conditions of single mineral systems are reported in many studies, there are still many points to investigate for mixed systems with close similarities in their physical and chemical properties [7–10]. Thus, according to its silicate content and similar properties to quartz, Swarna et al. [1], investigated the solubility of wollastonite and its Hallimond flotation behavior as a function of dodecyl amine concentration and pH value. Their results showed that above $5 \times 10^{-5}$ M amine concentration, a sudden increase in the hemimicelle concentration (HMC) was obtained for the flotation of wollastonite as a function of collector concentration [1]. It is worth noting that their finding was also in line with the flotation of quartz as a function of amine concentration which proved their similar physicochemical properties during flotation [11]. One another important finding was the final pH of

the suspension upon completing the dissolution of $Ca^{2+}$ ions to suspension became 9.5 in which the amine complexes on particle surfaces became maximum and effective on their flotation [12]. Apart from amine and derivatives of amine salts, the usage of fatty acids was also investigated for the separation of wollastonite from calcite [4]. Thus, selective flotation of wollastonite from calcite is a bit more difficult than other types of gangue minerals. This situation can be attributed to two main reasons; i) as wollastonite presents a needle-shaped crystal structure, it deteriorates the attachment of particles on bubbles during flotation and remains in the tailings section, ii) depending on similar color of wollastonite and enrichment conditions of calcite, more attention is required to determine the optimum conditions for selective flotation [4]. In this manner, initial calcite flotation is conducted after conditioning with fatty acid, and silicate flotation is conducted by using a mixture of anionic and cationic collectors [13]. The gangue minerals contained in the ore have similar flotation properties as wollastonite and therefore, selective flotation of wollastonite from gangue minerals is rather difficult. Reverse gangue flotation or bulk flotation followed by separation of wollastonite from gangue minerals is often practiced. When the ore contains mostly calcite, the ground is first conditioned with fatty acid followed by calcite flotation [13]. However, in the literature, fundamental research on the selective separation of these two minerals in the carbonate and silicate form by using fatty acids is limited. Although some studies are present for the enrichment of wollastonite or alike minerals by flotation, few of them tend to investigate the physicochemical properties of wollastonite minerals in the mixture and correlate them with batch flotation data.

The aim of this study is to characterize the physicochemical properties of wollastonite and calcite by zeta potential and contact angle measurements along with their flotation characteristics under optimum conditions.

## 2. Materials and Methods

### 2.1. Materials

For experimental studies, ESAN Mining Company calcite and wollastonite samples, taken from quarries in Turkey with hand sorting, was reduced to under 100 µm and to determine the size distribution of samples, particle size measurements were carried out for each sample with a Malvern Mastersizer 2000 (Malvern Panalytical Ltd, Malvern, UK) (Figures 1 and 2).

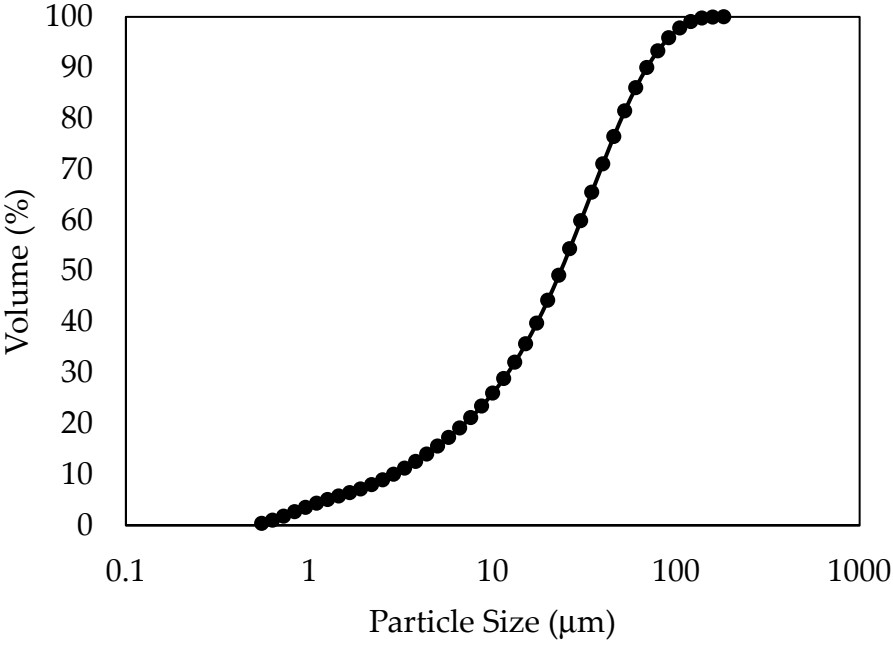

**Figure 1.** Particle size distribution of wollastonite.

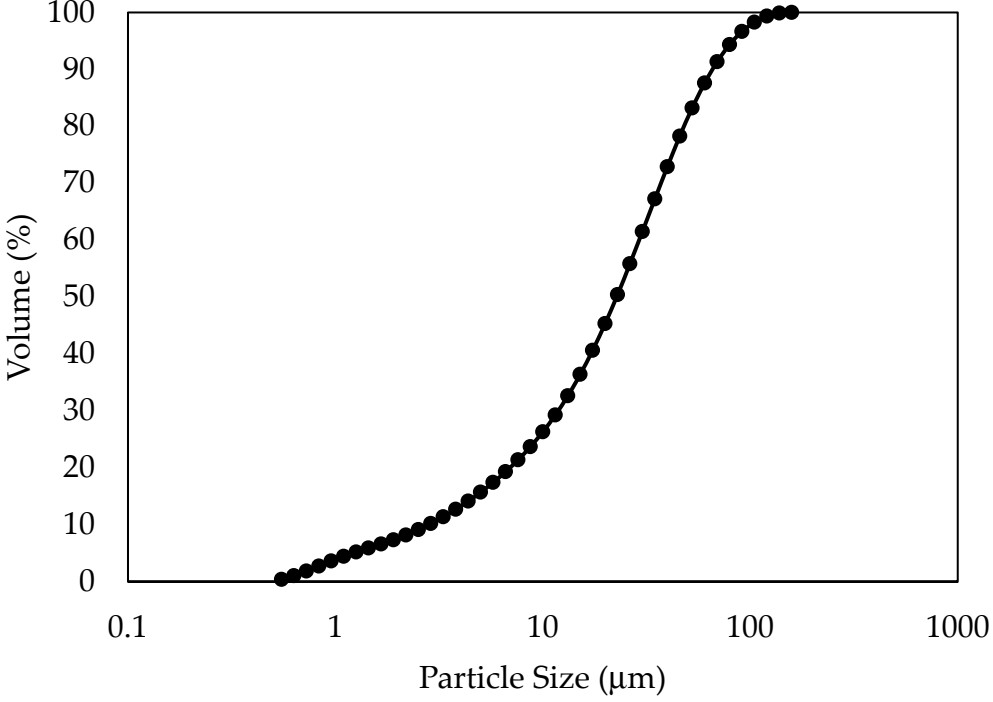

**Figure 2.** Particle size distribution of calcite.

As shown in Figures 1 and 2, while the $d_{90}$ of calcite was 58.6 μm, it is a bit finer for the wollastonite sample at 50.5 μm. In addition, the mineralogical and chemical composition of each sample was determined with X-Ray Diffraction Cu X-ray sourced Panalytical X'Pert Pro diffractometer (Malvern Panalytical Ltd, Malvern, UK) and X-Ray Fluorescence units, respectively. The mineralogical analysis of each mineral is shown in Figure 3. The chemical analysis results are shown in Table 1. Purities were approximately calculated by considering $SiO_2$ content for wollastonite as well as a loss on ignition for calcite because of the common calcium content of both wollastonite and calcite. For calculation of loss of ignition, samples were placed in weighed crucibles and weighed. Weight loss was measured after heating the samples overnight at 100 °C to remove water, at 550 °C for four hours to remove organic matter, and at 1000 °C for two hours to remove carbonates.

Based on analyses of both wollastonite and calcite samples, they were adequately pure enough for surface chemistry-based characterization. During characterization tests, potassium oleate (K-Ol) with a molecular weight of 320.55 g/mol specified by the manufacturer (Sigma Aldrich) was used as a collector reagent in the preparation of solutions with different molarity. The pH adjustments were made with dilute solutions of reagent grade HCl and NaOH. During all experiments, distilled water with 15 μmhos/cm S.C. (Specific Conductivity) was used to prevent the effect of other elements on measurements and flotation tests. In addition to water, technical grade hexane was used in contact angle measurements with the capillary rise method.

**Table 1.** Mineralogical composition of wollastonite and calcite minerals.

| Component (%) | Wollastonite | Calcite |
|---|---|---|
| $Fe_2O_3$ | 0.23 | 0.00 |
| $SiO_2$ | 50.94 | 0.00 |
| $Al_2O_3$ | 2.45 | 0.62 |
| CaO | 45.56 | 55.28 |
| Loss on Ignition | 0.52 | 42.05 |
| Purity | 84.90 | 98.71 |

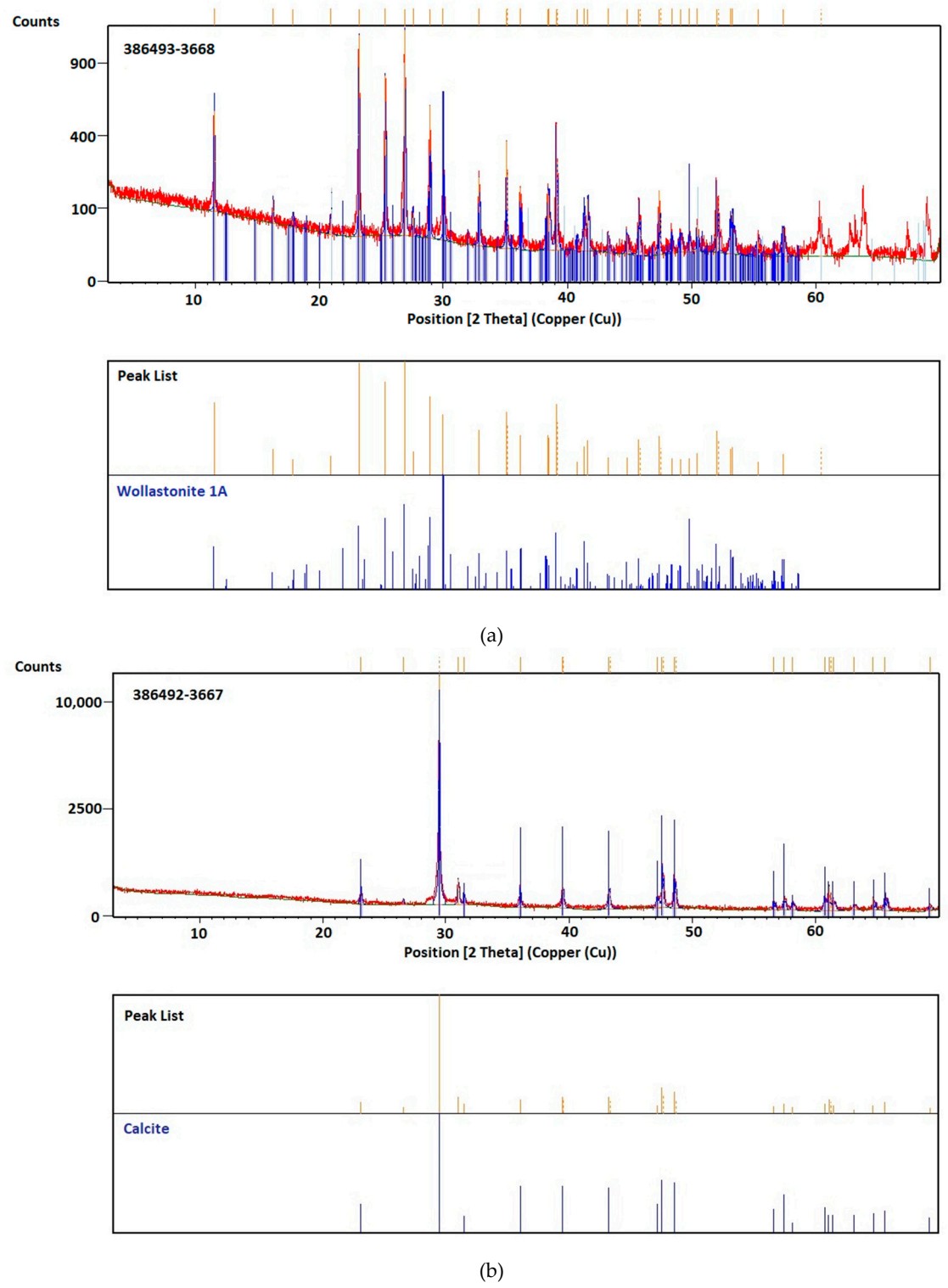

**Figure 3.** XRD analyses of wollastonite (**a**) and calcite (**b**) samples.

## 2.2. Methods

### 2.2.1. Surface Tension Measurements

Prior to characterization tests, the surface tension values of potassium-oleate reagent were determined as a function of molarity by Du-Noüy tension ring unit (Figure 4). A Du Noüy Ring Tensiometer (Krüss ®) (KRÜSS GmbH, Hamburg, Germany) was used to measure the surface tension

of K-Oleate solutions at varying concentrations under the original pH value (7.3 ± 0.02). In these tests, a mechanical tensiometer (Krüss K6) (KRÜSS GmbH, Hamburg, Germany) was used. The procedure during these tests is explained as follows: The maximum force that occurs upon moving of the platinum-iridium ring through the phase boundary is measured when the ring is aligned vertically to the ring plane. Thus, the surface tension is determined by stretching the lamella until detachment from liquid. A maximum force $F_{max}$ occurs when the lamella, which is produced when the ring moves through the phase boundary, is aligned vertically to the ring plane. This maximum correlates with the surface tension σ or interfacial tension according to Equation (1):

$$\sigma = \frac{F}{L \cdot cos\theta} \tag{1}$$

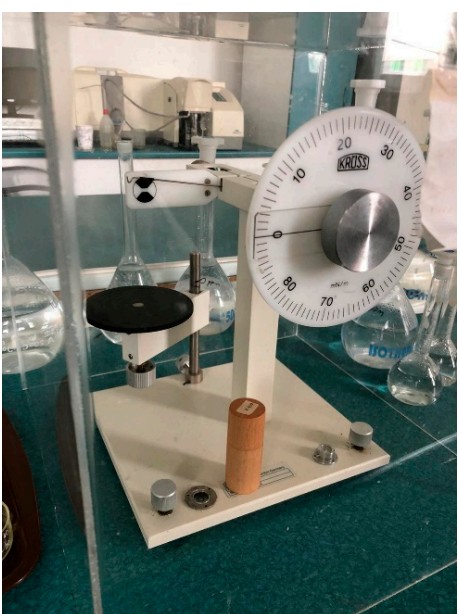

**Figure 4.** Du-Noüy surface tension ring unit.

All measurements were done in distilled water and at the original pH value of the sample (7.3 ± 0.02).

### 2.2.2. Zeta Potential Measurements

Zeta potential measurements were conducted with a microprocessor (ZM3-U-G, SOMATCO, Riyadh, Saudi Arabia) equipped Zeta-Meter 3.0+ model instrument. All measurements were carried out under 75 V and the K factor of measurement cells was 0.71 cm$^{-1}$. The sample for zeta potential measurements was prepared as follows: About a 0.5 g sample was added to 50 mL collector suspension at the desired concentration and mixed at 360 rpm for 15 min to provide suitable conditions for zeta potential measurements by considering the dissolution of Ca$^{2+}$ dissolution into suspension. This point is quite important for analyzing processes of salt type minerals because if the suspension has not reached the stable state, then the measured values will not present reliable results. All measurements were performed in 10$^{-3}$ M NaCl solution for obtaining equilibrium conditions for measurement, while diluted hydrochloric acid and sodium hydroxide were used as pH adjusters. This procedure assured the measurements under in-situ conditions. The initial and final pH values of the liquids were recorded for each sample. The average of at least ten measurements together with their standard deviations for each dispersion were recorded.

### 2.2.3. Contact Angle Measurements

Contact angle is a measure of interaction between solid and liquid interfaces. Determination of contact angle of a mineral or any other substance in a pre-defined suspension helps us to reach its wettability properties [14]. Although different methods are available for measuring the contact angle values of plates or plate-shaped materials, capillary rise and thin layer wicking methods are the ones for determination of the wettability of powder systems, which makes them important and usable for mineral processing [15–17].

In this study, the capillary rise method was performed for measuring the contact angle values of both calcite and wollastonite in the presence of potassium oleate solutions with different molarity. In this method, the increase in liquids with different polarity (water and hexane) were recorded as a function of time by the measurement of mass gain. The schematic presentation of the system is shown in Figure 5. As shown in the figure, samples were manually placed in a glass column with a 4 mm diameter and 10 cm height, which were closed by a non-woven fabric to support the bed. The decrease in mass of the container including polar (water) or nonpolar (hexane), was recorded every 5 s using an electronic balance. The time t = 0 approximately corresponded to the exact moment of the submersion of the column in wetting liquid. Thus, the procedure and the packing of particles play a very significant role in obtaining reliable results. The packing of particles was conducted under the same tapping time and number. The properties of liquids used in experiments are given in Table 2.

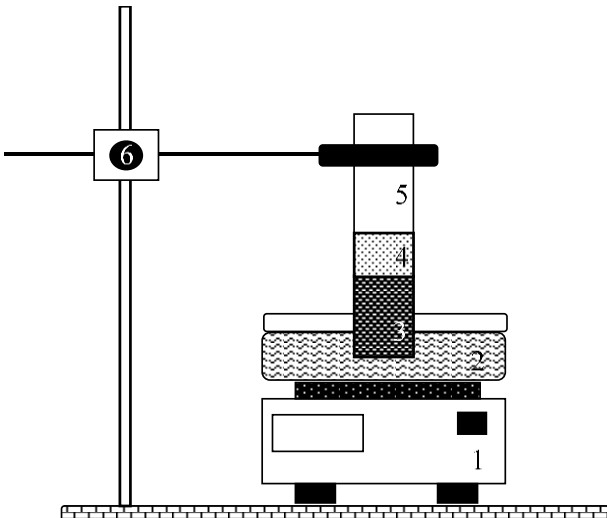

**Figure 5.** Laboratory set-up used for contact angle measurements by capillary rise method: 1—Electronic Balance, 2—Container with liquid, 3—Wetted sections of tapped solids, 4—Particle bed, 5—Column with 5 mm diameter, 6—Micrometric screw.

**Table 2.** Characteristics of liquids used for capillary rise experiments (20 °C).

| Wetting Liquid | Density (kg/m$^3$) | Viscosity (mPa·s) | Surface Tension (mJ/m$^2$) |
|:---:|:---:|:---:|:---:|
| Water | 997 | 0.00326 | 18.4 |
| Hexane | 655 | 0.01 | 72.8 |

Considering that restrictions on the calculation of contact angle and surface energy of minerals, the modified Washburn's equation involving the dependence of mass gain and time was used [18]. The relation between liquid mass and height in the column is given Equation (2):

$$m^2 = \frac{C\rho^2 \gamma \cos\theta}{\eta} \tag{2}$$

where $m$ denotes the mass of liquid, $C$ is the effective pore diameter, $\rho$ is the density, $\gamma$ is the surface tension, and $\eta$ is the viscosity of the liquid.

## 3. Results and Discussion

### 3.1. Surface Tension Measurements

The surface tension of potassium-oleate as a function of its concentration, M, was measured with the Du Noüy Ring method. As the pH value is one of the cornerstones for the evaluation of their effectivity, and since small changes in pH can significantly influence the surface tension of oleate solutions, it was used in its original pH to eliminate the contribution of any kind of hydrogen or hydroxyl species on surface tension measurements.

The results shown in Figure 6 indicated that, while the surface tension of potassium oleate was 65 dyn/cm at $2 \times 10^{-6}$ M concentration, it decreased to 52 when the concentration value increased to $2 \times 10^{-4}$ M.

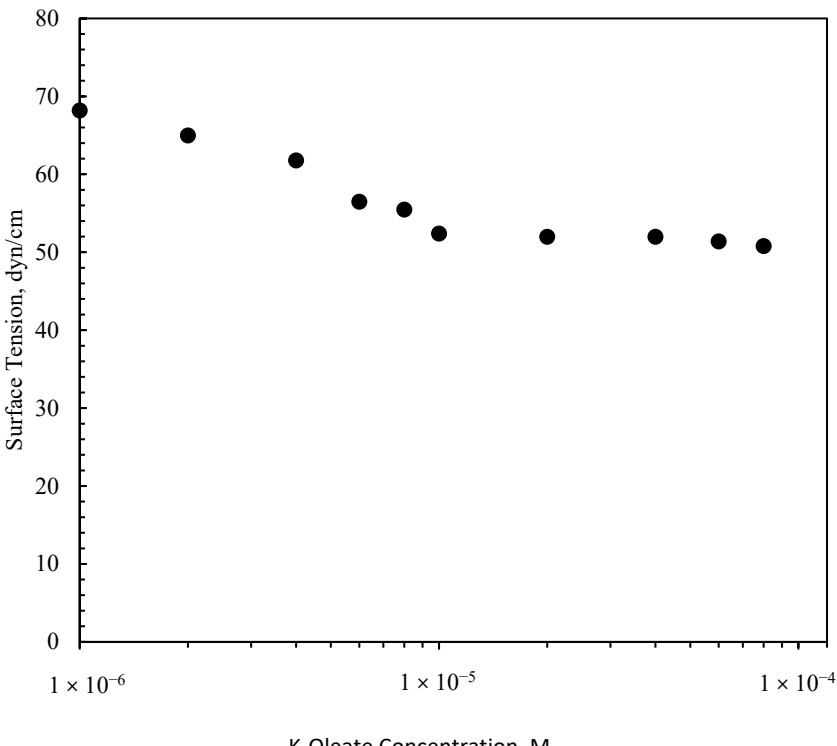

**Figure 6.** Surface tension measurements as a function of K-Oleate concentration.

### 3.2. Zeta Potential Measurements

As mentioned in the Methods section, the zeta potential of both calcite and wollastonite was conducted in the presence of potassium oleate, which was used during flotation tests. The results of these tests are shown in Figure 7.

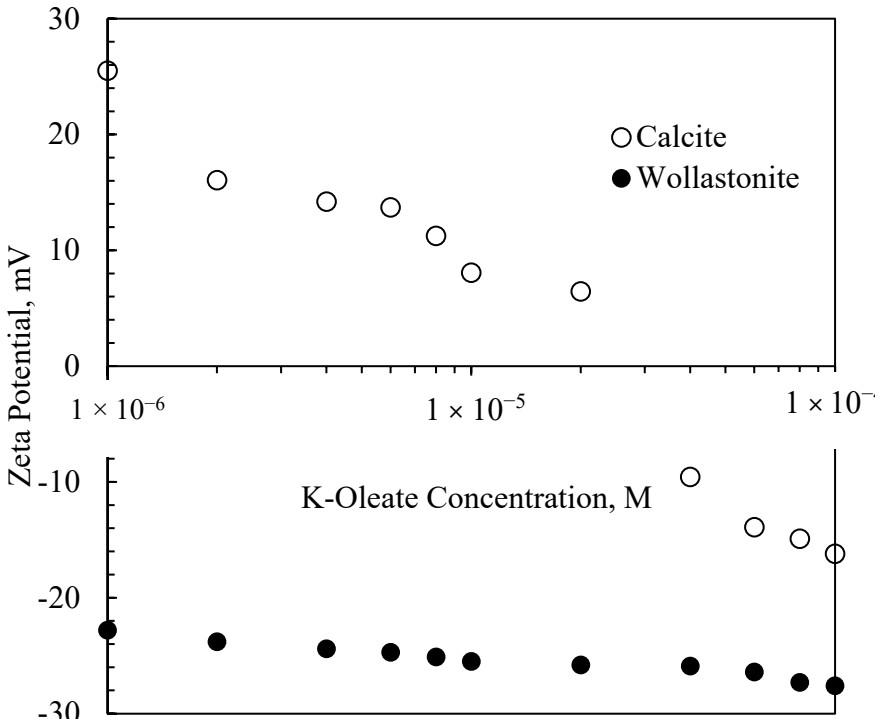

**Figure 7.** Zeta potential measurements of calcite and wollastonite.

As shown in Figure 7 that upon increasing the concentration of potassium oleate from $10^{-6}$ to $10^{-3}$ M, the zeta potential of calcite mineral varied from +25.51 mV to −16.2 mV. These results indicated the stepwise adsorption of oleate on calcite surfaces. In other words, upon the introduction of potassium oleate to this system, some of the oleate ions will precipitate as calcium oleate, while others remain as oleic acid and form a monolayer coverage on calcite surfaces. These results also showed the selective adsorption of potassium oleate on calcite under these conditions, which are in line with their flotation characteristics [4]. In addition to the effect of reagent concentration, El-Mofty and El-Midany, 2015 showed that when the potassium oleate concentration was $2 \times 10^{-4}$ M, the zeta potential value was measured in the positive region when the pH value was 7.7, it considerably decreased to negative values when the pH value was adjusted to 8.5 [19]. Therefore, besides other parameters, such as particle size, bubble size, mixing speed, the flotation of these minerals can be achieved after the determination of optimum reagent concentration and pH value.

On the other hand, only a negligible change was obtained for the zeta potential of wollastonite particles that can be attributed to the presence of silicate groups in its structure. Thus, in literature, Swarna et al. [1], used amine for flotation of wollastonite confirmed by dissolution, reagent adsorption, contact angle, and surface free energy measurements [1]. In these studies, it was found that, while the dissolution of calcium and silicate shows an opposite trend, the highest flotation recoveries were obtained at $1 \times 10^{-4}$ mol/L amine concentration, which was equivalent to a 50% monolayer coverage. It was also shown that maximum recovery could be obtained at pH 8.5–10.5, which was very similar to the flotation characteristics of other siliceous minerals, such as quartz and mica [14,20].

### 3.3. Contact Angle Measurements

As distinct from previous studies in literature, in this study, the contact angle values of wollastonite and calcite minerals in the presence of different potassium oleate concentrations were determined by the capillary rise method. The results of these tests are shown in Figure 8.

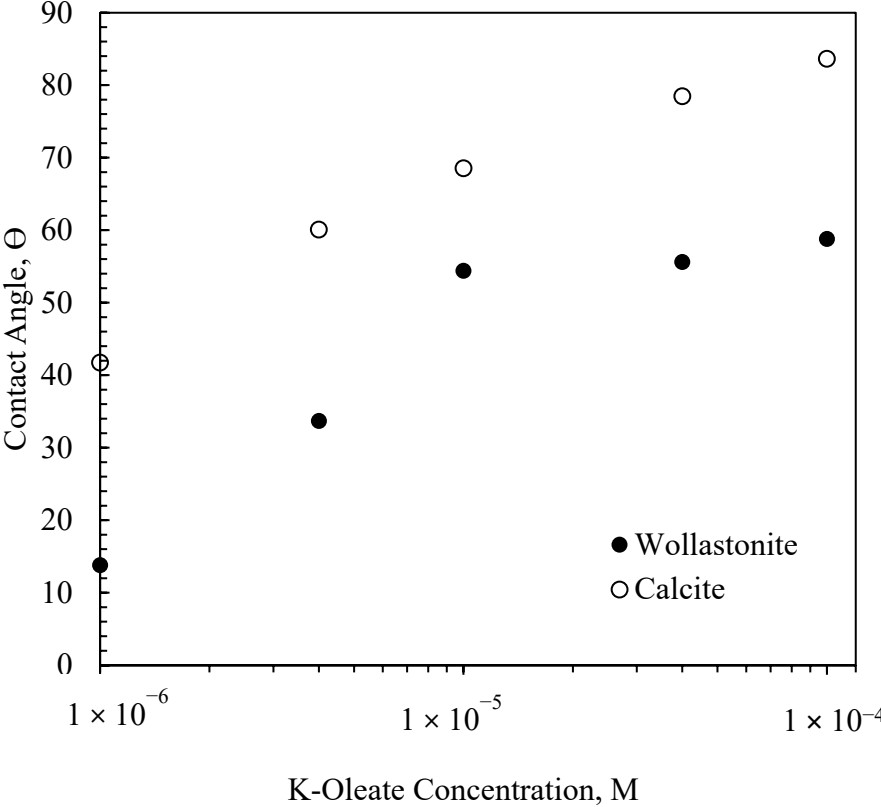

**Figure 8.** Contact angle measurements of calcite and wollastonite minerals as a function of potassium oleate concentration.

In Figure 8, while the contact angle values of calcite varied from 41.7 to 83.6 degrees, the variation for wollastonite was measured from 13.8 to 58.8 degrees. These values are in line with possible adsorption degrees of potassium oleate on calcite and wollastonite surfaces (retrieved from zeta potential values). Upon increasing oleate concentration in the system, a considerable increase was also obtained for contact angle values for each mineral. Although these values were contrary in some ways to their zeta potential trend, their flotation results proved this variation, such that when the recovery of only calcite mineral was 97.3%, it was found as 47.7% for wollastonite mineral [4]. Thus, there may also be some differences with other methods, which can be well ascribed to the rising characteristics of polar and non-polar solutions (water and hexane) on powder sized particle surfaces, unexpected free spaces between pressed particles, or wrong orientation of particles upon tapping.

*3.4. Correlation between Contact Angle and Zeta Potential Measurements*

To make a correlation between measurements, combined presentations of contact angle and zeta potential measurements are shown individually for each mineral (Figures 9 and 10).

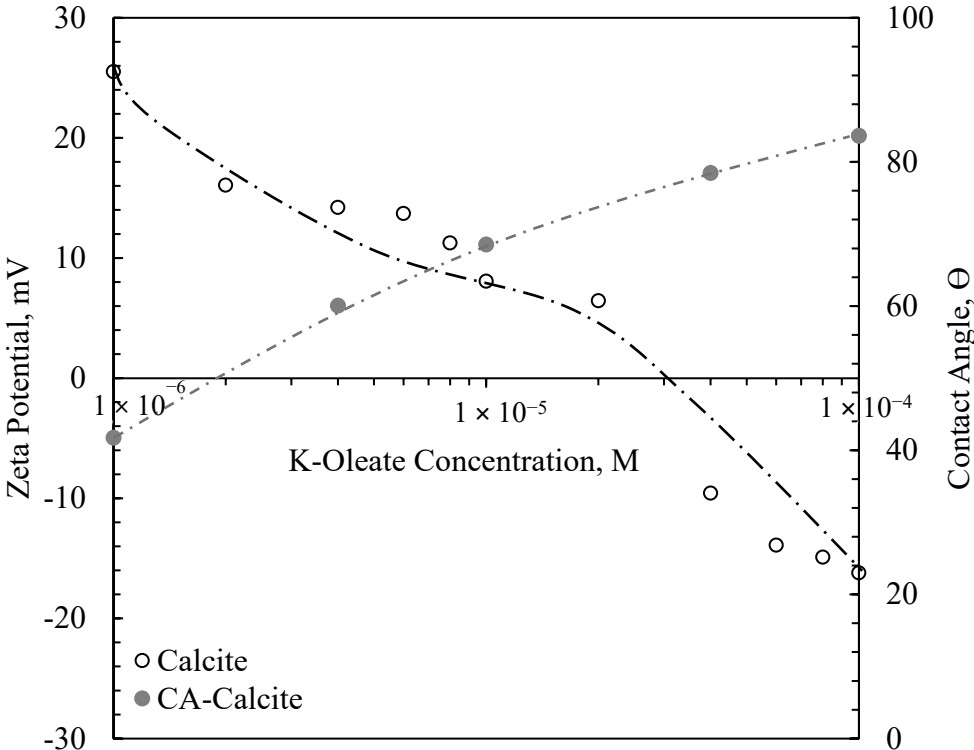

**Figure 9.** Combined presentation of contact angle and zeta potential measurements for calcite.

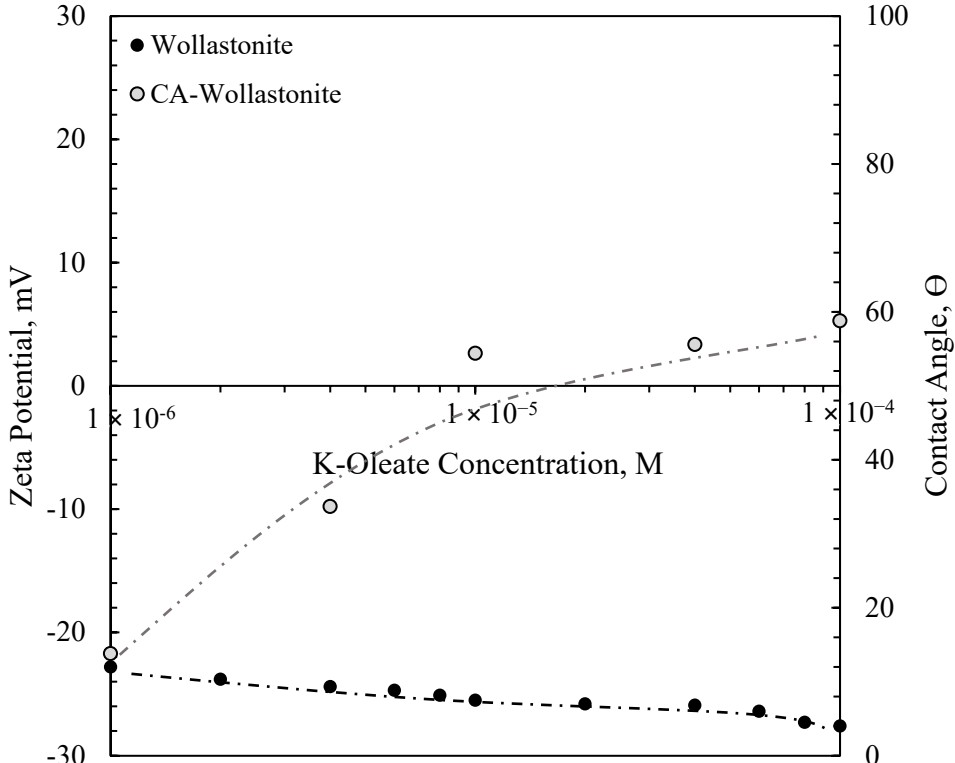

**Figure 10.** Combined presentation of contact angle and zeta potential measurements for wollastonite.

As shown in Figures 9 and 10, while a considerable increase was obtained in both contact angle and zeta potential values of calcite, it remained almost negligible in measurement for wollastonite. This trend then clearly showed that the adsorption rate of oleate on calcite surfaces was expected to be

higher than on wollastonite surfaces. As mentioned in previous sections, if a comparison was made for the floatability of each ore under the same conditions, the ultimate calcite concentrate obtained as another product from the experimental studies has a 99.80% calcite content and 85.4% recovery. A large loss on ignition (LOI) in wollastonite concentrate was obtained at pH 10 because calcite was floated well in the first two stages with potassium oleate, which is a fatty acid type collector [21]. On the other hand, pH 6 is suitable for activating iron-bearing minerals which are silicate type, such as augite. When all the experimental results were evaluated, the best result was obtained at pH 6, and 1500 g/t amount of K-Oleate. The calcite produced contained 55.89% CaO, 0.35% $SiO_2$, 0.03% $Fe_2O_3$, and 42.30% LOI, as well as a wollastonite concentrate, was obtained with 52.71% $SiO_2$, 44.65% CaO, 0.44% $Fe_2O_3$, and 0.60% LOI [4]. Thus, these findings also revealed that upon the usage of potassium oleate in the flotation of Ca-bearing wollastonite ore, it selectively floats calcite and wollastonite remained in the tailing section (Figure 11).

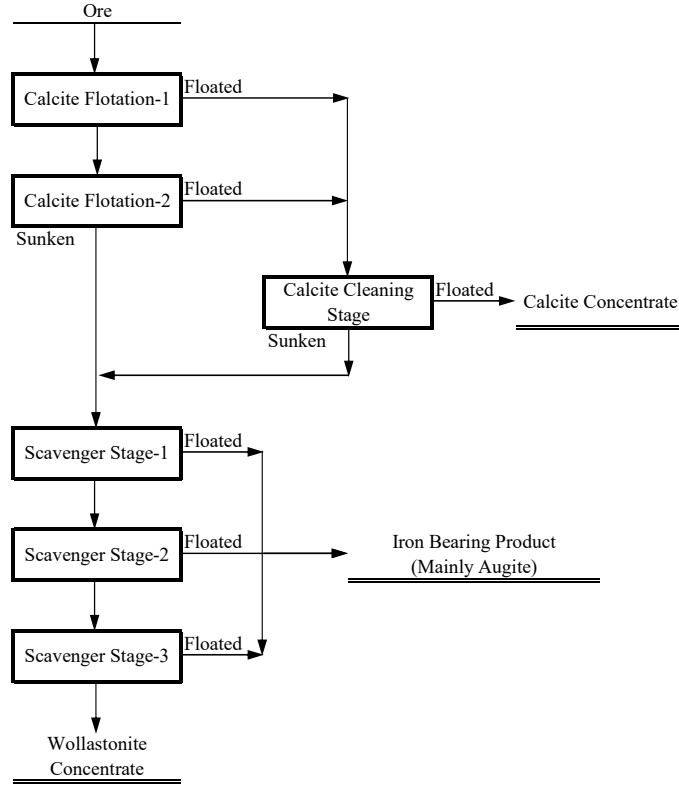

**Figure 11.** Flowsheet of the concentration of calcite and wollastonite [4].

## 4. Conclusions

Because wollastonite forms needle-like crystals, selective flotation of wollastonite from gangue minerals is rather difficult. Reverse gangue flotation or bulk flotation, followed by separation of wollastonite from gangue minerals, is often practiced [13]. As well-cited in previous reports for other types of minerals [14], the influence of shape factor, angularity in particular, not only changes the interaction between particles but also enhances the particle-bubble interactions, which, in turn, affects their flotation recoveries. Taking into account that knowledge, the flotation recoveries of wollastonite, which has a needle-like shape, is affected by this variable which is effective on adsorption of the collector on its surfaces and so its flotation recoveries vary.

The findings from this study provide a new light for considering the underlying mechanisms of Ca-bearing wollastonite flotation by evaluating the results of different measurements, such as contact angle and zeta potential values, under the same conditions. It was found that, while a negligible change was obtained on the zeta potential of wollastonite, a similar trend was found for its contact

angle values. In contrast to wollastonite, a significant increase was obtained for calcite both in its contact angle and zeta potential values under the same collector concentrations. This, in turn, revealed the selective adsorption of K-Oleate (a fatty acid derivative) on calcite, which explained the higher flotation recoveries of calcite and wollastonite in the same system.

**Author Contributions:** M.O.K., G.B. and O.G. conceived and designed the experiments; all researchers performed the experiments and analyzed the data; M.O.K., G.B. and O.G. wrote the paper. All authors have read and agreed to the published version of the manuscript.

**Funding:** This research received no external funding.

**Acknowledgments:** The authors would like to express their sincere thanks and appreciation to ESAN Mining Company for kindly providing the experimental samples.

**Conflicts of Interest:** The authors declare no conflict of interest.

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
