# Peer review of "Physicochemical Characterization of Natural Wollastonite and Calcite"

_minerals, doi:10.3390/min10030228_

Round 1

Reviewer 1 Report

Major comments

An important question. Is it a general physicochemical characterization or is it a case study?

Results need to be discussed in detail, please consider study of the correlation of the particle shape (wollastonite case) with their physicochemical parameters in flotation, i.e. needle like shape can influence the adsorption of k oleate on wollastonite surface

Minor comments

The title should contain some word that highlights the process of floating these mineralsIn figure 3, explanation for eser calcite, az quartz, az k-fel, and az dolomite should be stated

Figure 8 and 9 adjust scale so that the lines fit the graph

Author Response

We would like to thank reviewer for his/her valuable explanations and improvement’s on this paper. The editing job was completed according to comments and the answers are given below.

Major comments

An important question. Is it a general physicochemical characterization or is it a case study?

This study consists of physicochemical studies on both minerals. However, the results obtained are explained in a way to correlate with the enrichment studies performed before. Both studies consist of the data of a project. BULUT, G., AKÇÄ°N, E. S., KANGAL, M. O., (2019), “Value Added Industrial Minerals Production from Calcite-Rich Wollastonite”, Gospodarka Surowcami Mineralnymi–Mineral Resources Management, 35 (1), 43-58

Results need to be discussed in detail, please consider study of the correlation of the particle shape (wollastonite case) with their physicochemical parameters in flotation, i.e. needle like shape can influence the adsorption of k oleate on wollastonite surface

It was added.

Minor comments

The title should contain some word that highlights the process of floating these minerals. In figure 3, explanation for eser calcite, az quartz, az k-fel, and az dolomite should be stated

Abbreviations of minerals’peaks were forgotten on XRD curves. They were erased.

Figure 8 and 9 adjust scale so that the lines fit the graph

It was done.

Reviewer 2 Report

2.1 Materials.  You mention samples.  Where were the taken.  Initial state? Did you have to prep them?

Line 75 - "As shown in Figures.." what figures?

Line 77, what XRD machine was used and the software for deconvolution of the spectra.

Table 1 - mineralogical , not chemical composition (it's nto elemental)

Line 79. Don't start a sentence with Because please.

Line 80 LOI, can you explain the process please.

Figure 3 a & b, you need to expand the captions to separate them  I think.

Line 99  You need to explain the surface tension measurement  in this paragraph.  Do you have a schematic of methodology to elaborate upon?

Author Response

We would like to thank reviewer for his/her valuable explanations and improvement’s on this paper. The editing job was completed according to comments and the answers are given below.

2.1 Materials. You mention samples. Where were the taken. Initial state? Did you have to prep them?

It was added.

Line 75 - "As shown in Figures." what figures?

“Figures” was changed as Figures 1 and 2.

Line 77, what XRD machine was used and the software for deconvolution of the spectra.

Type of XRD machine was added.

Table 1 - mineralogical, not chemical composition (it's not elemental)

It was changed as “Mineralogical”.

Line 79. Don't start a sentence with Because please.

It was changed as “Purities were approximately calculated by considering SiO2 content for wollastonite as well as loss on ignition for calcite because of the common calcium content of both wollastonite and calcite.”

Line 80 LOI, can you explain the process please.

It was added.

Figure 3 a & b, you need to expand the captions to separate them I think.

Figures were partially expanded to fit the page structure

Line 99 You need to explain the surface tension measurement in this paragraph. Do you have a schematic of methodology to elaborate upon?

The surface tension measurement was explained. Figure of the surface tension unit was added.